# Fluorescent, Prussian Blue-Based Biocompatible Nanoparticle System for Multimodal Imaging Contrast

**DOI:** 10.3390/nano10091732

**Published:** 2020-08-31

**Authors:** László Forgách, Nikolett Hegedűs, Ildikó Horváth, Bálint Kiss, Noémi Kovács, Zoltán Varga, Géza Jakab, Tibor Kovács, Parasuraman Padmanabhan, Krisztián Szigeti, Domokos Máthé

**Affiliations:** 1Department of Biophysics and Radiation Biology, Semmelweis University, 1085 Budapest, Hungary; hegedus.nikolett@med.semmelweis-univ.hu (N.H.); horvath.ildiko@med.semmelweis-univ.hu (I.H.); kiss.balint@med.semmelweis-univ.hu (B.K.); kovacsnoi@hotmail.com (N.K.); varga.zoltan@ttk.mta.hu (Z.V.); 2Institute of Materials and Environmental Chemistry, Research Centre for Natural Sciences, 1117 Budapest, Hungary; 3Department of Pharmaceutics, Semmelweis University, 1085 Budapest, Hungary; jakab.geza@pharma.semmelweis-univ.hu; 4Institute of Radiochemistry and Radioecology, University of Pannonia, 8200 Veszprém, Hungary; kt@almos.vein.hu; 5Lee Kong Chian School of Medicine, Nanyang Technological University, Singapore 636921, Singapore; ppadmanabhan@ntu.edu.sg; 6In Vivo Imaging Advanced Core Facility, Hungarian Centre of Excellence for Molecular Medicine, 6723 Szeged, Hungary; 7CROmed Translational Research Centers, 1047 Budapest, Hungary

**Keywords:** Prussian blue nanoparticles, fluorescent imaging, optical imaging, biocompatible

## Abstract

(1) Background. The main goal of this work was to develop a fluorescent dye-labelling technique for our previously described nanosized platform, citrate-coated Prussian blue (PB) nanoparticles (PBNPs). In addition, characteristics and stability of the PB nanoparticles labelled with fluorescent dyes were determined. (2) Methods. We adsorbed the fluorescent dyes Eosin Y and Rhodamine B and methylene blue (MB) to PB-nanoparticle systems. The physicochemical properties of these fluorescent dye-labeled PBNPs (iron(II);iron(III);octadecacyanide) were determined using atomic force microscopy, dynamic light scattering, zeta potential measurements, scanning- and transmission electron microscopy, X-ray diffraction, and Fourier-transformation infrared spectroscopy. A methylene-blue (MB) labelled, polyethylene-glycol stabilized PBNP platform was selected for further assessment of in vivo distribution and fluorescent imaging after intravenous administration in mice. (3) Results. The MB-labelled particles emitted a strong fluorescent signal at 662 nm. We found that the fluorescent light emission and steric stabilization made this PBNP-MB particle platform applicable for in vivo optical imaging. (4) Conclusion. We successfully produced a fluorescent and stable, Prussian blue-based nanosystem. The particles can be used as a platform for imaging contrast enhancement. In vivo stability and biodistribution studies revealed new aspects of the use of PBNPs.

## 1. Introduction

Optical methods are simple and easy ways of imaging in biomedical research. They offer opportunities from simple observation of subjects to advanced methods for the imaging of tumors and different metabolic pathways using different kinds of spectroscopy and microscopy [1]. The major drawback of this method lies in the properties of light: tissue absorption and scatter strongly depend on the wavelength of light as well as the tissues themselves. Strong autofluorescence of the highly perfused tissues, e.g., muscles, skin, and hair scatter limit the tissue penetration and contrast of optical methods. Fluorophores, emitting light at higher wavelengths can potentially result in a better signal-to-noise ratio and a better image [2,3,4]. On the other hand, fluorophores alone are not able to mark specific tissues; their premature metabolism after intravenous administration often leads to decreased uptake in the target tissue or tumor. Yet, their toxicity to the healthy tissues is another issue. To overcome these drawbacks, chemical or physicochemical solutions would connect these NIR fluorophores to other structures, e.g., nanoparticles or proteins, which could solve the toxicity and sensitivity issues, allowing the microdosing of these agents [4,5].

Mesoporous nanoparticles have been reported, capable of fluorescent optical imaging [6,7,8,9]. The mesoporous structure itself could lend the particle absorbing properties of ions, anticancer drug, or even fluorescent dyes to the corresponding particles [6,7,8,9,10]. However, the synthesis of such particles can range from a very wide range of chemical synthesis techniques from the polycondensation process to thermal oxidation to thermal condensation; the basics of nanoparticle separation via centrifugation, rinsing the samples using ethanol to remove “unwanted,” unreacted contaminants along with dialyzing the samples could be easily translated and implemented into any synthesis process [6,9].

Methylene blue, with its excitation wavelength at 633 nm and emission at 662 nm could be a potential candidate for fluorescent imaging. Its emission wavelength is in the diagnostic window (650–900 nm), where the extinction coefficients of oxyhemoglobin, deoxyhemoglobin, and water are the lowest [4]. Furthermore, methylene blue is on the World Health Organization List of Essential Medicines, the safest and most effective medicines needed in a health system and has been used widely in clinical and basic research [10,11]. Its properties and widespread global use make it a promising candidate for the fluorescent labelling of a multimodal imaging system, a Prussian blue nanoparticle-based nanoteragnostic agent (Prussian blue (PB) nanoparticles (PBNPs)).

Previous studies prove that PBNPs could possibly act as zeolitic and porous structures, absorbing metallic cations and gases in their pores, although the exact binding mechanisms were not fully described and published [12,13,14]. It was also previously reported that PBNPs could have a great potential as multimodal contrast agents as well as potential drugs for photodynamic therapy [7,13,15,16]. Equipped with a suitable biocompatible citrate layer would help to stabilize the particles and render them capable to form surface dendrimers when polyethylene glycols are added to the PBNP solution [7,12]. Just like with PEGylation, these surface dendrimers would prevent the aggregation of the nanosystem and provide long-term stability in physiological conditions [17,18].

Our main goal was to create a fluorescently labeled Prussian blue nanoparticle (PBNP) (iron(II);iron(III);octadecacyanide)-based nanosystem for preclinical fluorescent imaging. The crucial quality of the platform should be as follows: first, it should be a stable nanosystem, which does not aggregate over long periods of time; second, the emission of a fluorescent signal that could be detected in living organisms to make the particle suitable for in vivo imaging; and third, a clearance mechanism to reduce possible side effects but also to slow down accumulation in the monocyte-macrophage system, the liver and spleen. We wished to also take advantage of the biocompatible coating of the particle, thus connecting the potential fluorescent dyes and stabilizers to PBNPs. To this end, we selected and tested the fluorescent dyes eosin Y, rhodamine B, and methylene blue for further sorption to the PB nanoparticles, both with and without further (or preceding) PEGylation. We selected a 3-h imaging time point evaluation in order to model a clinically also relevant situation as in the case of currently running image-guided surgery clinical studies of fluorescent contrast materials.

## 2. Materials and Methods

### 2.1. Synthesis of PBNPs

Citrate-coated PBNPs were synthesized by following the method published by Shokouhimehr and colleagues with some modifications [14]. Briefly, PBNPs were prepared by slowly adding 20 mL of 1.0 mM Fe(III) chloride anhydrous (FeCl_3_; Sigma-Aldrich, Budapest, Hungary) solution with 0.5 mmol of citric acid (Sigma-Aldrich) to a solution containing an equimolar amount of potassium ferrocyanide anhydrous (K4[Fe(CN)6]; Sigma-Aldrich, Budapest, Hungary) under vigorous stirring for 15 min at 40 °C. The products were isolated with centrifugation at 21,130 rcf (Eppendorf 5424R centrifuge) at 4 °C.

### 2.2. Fluorescent Labelling

Eosin Y and rhodamine B were adsorbed to the particles as reported in Appendix A. For fluorescent labelling, the concentrated methylene blue (MB; Sigma-Aldrich) stock solution was diluted twofold and filtered through a 0.22 µm membrane filter (MILLEX GP 0.22 µm; Merck Millipore Ltd., Budapest, Hungary). Further, 200 µL of this filtered solution was added to 2 mL of the PBNP solution. This would result in absorbed MB on the mesoporous surface of the biocompatible PBNPs. After that, two main approaches were made:The synthesized PBNPs were labelled after centrifugation of “bare” PBNPs at 4 °C, using 21,130 rcf. Thereafter, the PEGylation process was performed and the solution was dialyzed against phosphate buffer saline solution (pH = 6.8; Ph.Eur. 8.).The PBNP-MB mixture was centrifuged at 21,130 rcf (Eppendorf 5424R centrifuge) at 4 °C. Then, the PBNP-MBs were PEGylated, dialyzed, and stored at 2–8 °C until further use.

### 2.3. PEGylation of PBNPs

For the PEGylation, PEG 3000 (for molecular biology; Sigma Aldrich, Darmstadt, Germany) was available in monodisperse solution, while PEG 6000 (for synthesis) and PEG 8000 (for synthesisUSP) (Sigma Aldrich, Budapest, Hungary) were commercially sold in solid form. At first, these power-based agents were dissolved in 50% ethanol–water mixture the final PEG content was set to 10 *w*/*w*%.

PBNP-MB solutions (2 mg/mL) were prepared by adding distilled water (Milli-Q) to the stock MB-labelled PBNP solution. After 15 min incubation time at room temperature, different PEG solutions (PEG 3000, PEG 6000, and PEG 8000) were added to the PBNP-MB solutions and dialyzed for 24 h (14 kDa filter) (Sigma-Aldrich, Budapest, Hungary) in phosphate buffer saline solution (pH = 6.8; Ph.Eur. 8.). Different *v*/*v*% concentration compositions (PEG 3000: 1.47-1.96-2.44-3.85-9.09, PEG 6000: 1.47-1.96-2.44-3.85, and PEG 8000: 1.47-1.96-2.44) were prepared and characterized in the following with DLS and Zetasizer instruments. Other details are available in the Appendix A.

### 2.4. DLS and Zeta Measurements

The absorbance of the samples was measured at 633 nm with a Model 8453 UV–VIS spectrophotometer (Agilent Technologies, Santa Clara, CA, USA). The surface charge and hydrodynamic diameter of the particles were determined using a Zetasizer Nano ZS (Malvern Instruments Ltd., Worcestershire, UK). DLS measurement was performed at 37 °C in backscattering mode (detector fixed at 173°) using a 633 nm He-Ne laser. Samples were measured in a reusable quartz cuvette (type PCS1115 6G; Malvern Instruments Ltd., Worcestershire, UK). Measurement of zeta potential was performed under similar conditions. DTS1070 disposable cuvettes were used (Malvern Instruments Ltd., Worcestershire, UK). The measurement data were evaluated using software provided by the manufacturer, and statistical data and graphs were created and evaluated with Origin 9.0 (OriginLab) and Microsoft Excel 2013 software. DLS measurements were performed weekly for a period of 6 weeks to determine colloidal stability. Samples were stored at 2–8 °C.

### 2.5. Atomic Force Microscopy (AFM)

After a 10 min incubation time, mica surface was dried in N2 stream. AFM images were collected in noncontact mode with a Cypher S instrument (Asylum Research, Santa Barbara, CA, USA) at 1 Hz line-scanning rate in air, using a silicon cantilever (OMCL AC-160TS, Olympus, Tokyo, Japan) oscillated at its resonance frequency (300–320 MHz). Temperature during the measurements was 25 ± 1 °C. AFM amplitude-contrast images are shown in this paper. The filter used on the images enhances the details of the height contrast images (violet, orange, yellow). AFM images were analyzed by using the built-in algorithms of the AFM driver software (Igor Pro, Wave Metrics Inc., Lake Oswego, OR, USA). Particle statistics was done by analyzing a 4 μm × 4 μm height-contrast image with (*n* = 95) particles. Maximum height values were taken as the height of particles, and rectangularity was calculated as the ratio of the particle area to the area of a nonrotated inscribing rectangle. The closer a particle is to a rectangle, the closer this value is to unity. Detailed data is available in Appendix A.

### 2.6. Scanning Electron Microscopy (SEM)

The nanoparticle suspension was diluted with distilled water (1:2) and applied to a metallic sample plate, which was covered with a double-sided carbon tape. The sample was dried under vacuum, metallized with gold and investigated with a field emission EM, JEOL JSM 6380LA SEM (Jeol Ltd., Tokyo, Japan), at 15 kV and a working distance of 15 mm. The morphology of the nanoparticles was observed.

### 2.7. Transmission Electron Microscopy (TEM)

Morphological investigations of the NPs were carried out on a JEOL JEM 1200EX (Jeol Ltd., Tokyo, Japan) transmission electron microscope. The diluted sample was dropped and dried on a carbon-coated copper grid. The length (along multiple axles) and the area of the particles were determined by manually measuring the diameter (*n* = 304) and circumference (*n* = 176) of the nanoparticles, using ImageJ software.

### 2.8. Animals

In vivo fluorescent imaging tests of MB-PBNP nanosystems were carried out in C57BL/6 male mice (Janvier, France). Animals had ad libitum access to food and water and were housed under temperature-, humidity-, and light-controlled conditions. All procedures were conducted in accordance with the ARRIVE guidelines and the guidelines set by the European Communities Council Directive (86/609 EEC) and approved by the Animal Care and Use Committee of Semmelweis University (protocol number: XIV-I-001/29-7/2012). Mice were 10–12 weeks old with an average body weight of 28 ± 5 g. During imaging, animals were kept under anesthesia using a mixture of 2.5% isoflurane gas and medical oxygen. Their body temperature was maintained at 37 °C throughout imaging. For the most humane termination of the animals, intravenous Euthasol (pentobarbital/phenytoin) injection was used.

### 2.9. In Vivo Fluorescent Imaging

The fluorescent labelled PBNPs were imaged using a two-dimensional epifluorescent optical imaging instrument. (FOBITM, Fluorescent Organism Bioimaging Instrument; Neoscience Co., Ltd., Suwon-si, Korea). Mice were anesthetized with isoflurane (5% for induction and 1.5–2% to maintain the appropriate level of anesthesia; Baxter, AErrane). Precisely, 200 µL of MB-labelled PBNP solution containing 224 µg of Fe(III) in a 2 mg/mL concentration PBNP solution was administered intravenously into the tail vein. The images were collected (after shaving to remove hair) at 4 different time points (pre- and postinjection, 3 h postinjection, and ex vivo post the 3 h in vivo time) with excitation of 630–680 nm corresponding to the excitation maximum of the dye (ex: 664 nm; em: 687 nm). The emission spectrum of the dye was in the pass band of the used emission filter. Image acquisition parameters were the following: exposure time: 1000 msec and gain: 1. The images were evaluated with VivoQuant software (Invicro, 27 Drydock Avenue, Boston, MA, USA).

## 3. Results

### 3.1. Preformulation of PEGylated PBNPs

Multiple approaches were made to determine the optimal PEG coating concentration. For that, different volume-percent compositions were formed and characterized. During the preformulation, DLS measurements were performed and combined with the macroscopic analysis of the samples (aggregation, visual evaluation, homogeneity, and homogenizability) (Appendix A).

We found that the optimal PEG concentration for the different types of PEG solutions is slightly different if the particle size and the PDI are the form factors for choice. For further inspections, PBNP-MB samples containing 5 mg (10 μL) PEG 3000 (PNBP-MB@PEG3000_10μL), 10 mg (100 μL) PEG 6000 (PNBP-MB@PEG6000_100μL), and 12.5 mg (100 μL) PEG 8000 (PNBP-MB@PEG8000_100μL) were chosen after DLS measurements and macroscopic inspections. The other samples showed either aggregation during the measurements or sedimented despite the optimal hydrodynamic diameter and low PDI value. The synthesized methylene-blue-labelled PEGylated PBNP-s (PBNP-MB@PEG) were stored at 2–8 °C.

### 3.2. In Vitro Measurements

#### 3.2.1. DLS and Zeta Potential of Fluorescently Labelled PBNPs

MB labelled, PEGylated PBNPs were measured first to determine the optimal size. These measurements were part of a preformulation study, which included mainly macroscopic inspections along with DLS and zeta potential measurements (Appendix A).

#### 3.2.2. Atomic Force Microscopy (AFM)

AFM images are shown in Figure 1. PBNPs appeared as objects with a flat rectangular surface protruding from a rounded halo (Figure 1a). The rectangular surface represents the real geometry of the particles (Figure 1c) while their halo is the consequence of tip convolution, i.e., the effect of imaging a rectangular prism by a tetrahedral AFM tip. Rectangularity of the particles (together with their halo) was found to be 0.7701 ± 0.1041 (mean ± SD), indicating that PBNPs indeed represent rectangular topography. The height of the particles showed monomodal distribution with a mean ± SD of 17.790 ± 8.922 nm (Figure 1).

#### 3.2.3. Scanning Electron Microscopy (SEM) and Transmission Electron Microscopy

Due to the sample preparation process, SEM images show an aggregated nanosystem where the particles appear as rectangular objects. Single particles were hard to distinguish from one another, however, their size did not exceed the 100 nm threshold (Figure 2a).

The nonhydrated shape and size of the NPs were also analyzed with TEM (Figure 2b,c). The nanosystem was slightly aggregated (Figure 2c), however, single particles could be observed. The shape of the nanoparticles on TEM and AFM images was similar. PBNPs appeared as flat rectangular, dense objects in this case as well. The average length of the particles was 28.080 ± 6.690 (mean ± SD; *n* = 304) (Figure 2d), along with an average surface area of 582.20 ± 269.750 nm^2^ (mean ± SD; *n* = 176 particles) (Figure 2e).

#### 3.2.4. Measuring In Vitro Fluorescence

After the two main approaches of fluorescent labelling, PBNP-MB@PEGs were tested in vitro, to study whether the emitted signal is adequate for in vivo imaging. For that purpose, 20-fold diluted samples were produced and measured, along with the stock PBNP-MB@PEG solutions.

The samples prepared according to method a showed lower intensity on the emitted light, on the other hand, method b provided a significantly better signal in emission, which made the sample a better choice for in vivo use (Figure 3a,b).

#### 3.2.5. Stability Measurements of PBNP-MB@PEG Nanoparticles

The mean zeta potential of the measured samples (PBNP, PBNP-MB@PEG3000, PBNP-MB@PEG6000, and PBNP-MB@PEG8000) was not bigger than −20 mV, moreover, all the samples had a net negative charge of −25 mV or greater.

The mean hydrodynamic diameter of the particles was measured as a part of a 4-week stability test. However, non-modified PBNPs size showed a slight change in the diameter, i.e., after the third week, all PEGylated particles started to aggregate. PBNP-MB@PEG6000 nanoparticles were the most stable, along with the lowest PDI and diameter. As the size (hydrodynamic diameter) of the other two particles (PBNP-MB@PEG3000 and PBNP-MB@PEG8000) doubled after 3 weeks, the PBNP-MB@PEG6000 particles kept a solid size of 24.82 ± 5.83 nm (number-based particle size distribution). According to these results, PBNP-MB@PEG6000 particles remained stable for the longest time, thus made them the best candidate for further inspections (Appendix A)

### 3.3. In Vivo and Ex Vivo Measurements

Due to the strong visible fluorescent signal of PBNP-MB@PEG6000s, 200 μL of the suspension of nanoparticles was injected into the lateral tail vein of C57BL/6 male mice Figure 4A,B. The semiquantitative distribution of the labelled PBNPs was determined based on their normalized mean fluorescent intensity (Appendix A). On the ex vivo images Figure 4B, taken 3 h after the first injection, PBNP-MB@PEG6000 accumulation can be observed in the gastrointestinal tract, kidneys, spleen, liver, and heart.

## 4. Discussion

It has long been known that Prussian blue (PB) is capable of chelating toxic metals, e.g., cesium (Cs) and thallium (Tl), thus facilitating their clearance from the body. This property is the main reason PB is featured on the World Health Organization (WHO) Model List of Essential Medicines, as the specific category “Antidotes and other substances used in poisoning” PB has been approved for medical use in the United States (US) and in Europe for the abovementioned indication since 2003 [10,19]. Previous studies also described Prussian blue nanoparticles (PBNPs) as potential photothermal therapeutic agents [16], a promising candidate for different types of imaging [20], e.g., magnetic resonance-imaging (MRI) (as a T1 and T2 contrast agent) [12,21], optical imaging, and computed tomography [22]. The most widely used imaging modalities for follow-up studies are MRI and optical imaging (OI). MRI, based on the different relaxivities modulated by contrast agents (CA), is able to emphasize the structural (anatomical) differences between different types of tissues. MRI methods are noninvasive and can be easily translated from preclinical research to clinical applications, allowing the acquisition of spatial as well as temporal information. However, MRI suffers from less than optimum sensitivity of detection of injected CAs [23].

Of the mentioned modalities, OI is, however, the least complicated and most versatile method in current preclinical imaging applications. OI is free from harmful effects of ionizing radiation—in fact, it replaces it with nonionizing radiation, which includes visible, ultraviolet, and infrared light [24].

Several approaches were made to synthesize multimodal nanoparticles, which were capable of optical imaging along with other imaging techniques or had notable therapeutic relevance. Dumont et al. described PBNP analogue Mn (II) containing fluorescent-labelled nanoparticles with pediatric brain tumor (PBT) specific antigens. These nanoparticles generated MRI contrast (in both T1 and T2) and showed specific uptake in PBTs due to their surface modifications. These modifications granted them biocompatibility and provided fluorescent signal in vivo [25].

Tu et al. has also described multimodal magnetic-resonance and optical-imaging contrast agent, which were sensitive to NADH. In the presence of NADH, the MRI signal enhanced, whereas the green fluorescence disappeared [26]. The contrast agent itself had a notable MR signal enhancement in vivo, the only drawback of its clinical application was the lack of specificity on a cellular level. Furthermore, the absorption peak of the molecule was found in the region of autofluorescence, which made it a less promising candidate for further use.

To overcome the weakness in OI, Shcherbo et al., in his work, has discussed a protein known as mKate/Katushka (monomer/polymer), which could be excited at 588 nm and emits at 635 nm. At the far-red emission wavelengths, neither the absorbance of hemoglobin and melanin at 620 nm nor the absorbance of water at 1100 nm is significant, thereby providing greater penetration depth for detection in vivo [27]. Due to the lack of selectivity in fluorescent signal, this technique could not succeed in preclinical drug research, notwithstanding its novelty in whole-body imaging and gene manipulation.

Despite other important results, Prussian blue nanoparticles were not yet been described as both fluorescent and PEGylated nanoscale platform for in vivo imaging. In our work, the fluorescence was provided with an adsorbed dye, namely, methylene blue (MB), which is a highly water-soluble fluorescent dye. MB is widely used in different fields of medicine: as the medication and treatment of methemoglobinemia and cyanide poisoning; as a part of combination of drugs in infection control, which covers the urinary tract; and as a strain or dye in different types of endoscopy [28,29,30,31].

In addition, the excitation of methylene blue is in the NIR region (above 630 nm). MB is also the most inexpensive of the commercially available NIR fluorescent dyes and has been widely used for bioanalysis [32]. Prussian blue nanoparticles were previously described as biosensors, based on their redox properties. Their peroxidase-like activity was in focus of the studies of Zhang et al. and Vázquez-González et al. Their studies suggest that the interaction of cells with PBPNs could enhance the effects of oxidative damage to cells, especially to the cells of the liver, which was observed in the work of Chen et al. [33,34,35]. To overcome the oxidative effects of PBNP administration, either reductive substance should be administered with the PBNPs or reductase-like activity should be attached to the surface of the PBNPs. Such reductive substances would prevent the effects of the PBNPs. MB is a strong antioxidant, which would prevent cellular oxidation damage as Zhang et al. describes, by preventing mitochondrial oxygen free radical formation and enhancing oxygen consumption [36]. Previous studies described MB as an encapsulated dye in different kinds of particles for fluorescent imaging [37,38]. In addition, Szigeti et al. and Chen et al. suggested that mesoporous PBNPs could act as “chemical sponges,” absorbing metal ions and drugs, depending on the pH conditions and properties of the PBNPs [13,39] Our results suggest similar mechanisms for labelling. The two applied methods were mainly different: method a suggested a connection based on surface charges and secondary bonds, where PEG had an additional function as sealant, preventing MB leakage. In contrast, method b showed less fluorescence, which suggest less adsorbed dye and therefore, poor labelling efficacy. The surface of PBNPs was “sealed” before MB could connect to the particles, which resulted in increased dye-washout during dialysis.

Surface charge measurements have been concluded—they suggest a small decrease in the zeta potential after MB labelling (mean value before labelling <−30 mV; mean value after the labelling <−25 mV). In addition, additional surface charge measurements were concluded, on older samples. Their zeta potential did not exceed −20 mV. Therefore, it is possible that the decreased surface charge is the result of the ionic interactions between the dye and the citrate surface. Nevertheless, it must not be forgotten, that the pH of the used buffer (Phosphate buffer saline pH 6.8) could also result a decreased zeta potential in the MB-labelled PBNP sample. Previous studies described citric acid-, PVP-, and PEG-coated PBNPs. These capping agents are biocompatible and act mainly as steric stabilizers, which enhance the colloidal stability of the particles and affect the size and shape of the nanoparticles [40,41,42].

Chelating properties of citric acid were previously reported; citrate forms complexes with metallic cations although, at physiological conditions, these complexes have no effect on the colloidal stability of the particles [43,44]. Both citric acid and PEG offer PBNPs biocompatibility, moreover, citrate as a capping agent could function as an anchor for different molecules due to its terminal carboxylic function. PEG and citric acid were reported to form aggregates, connected with π-bonds, depending on the concentration and the proportion of citric acid to PEG [17,18].

The reported PEGylated MB-labelled PBNPs remained stable for about 3 weeks, while the previously reported, citrate-coated PBNPs did not aggregate in a biologically relevant manner after 4 weeks [9,18]. This lets us conclude that the increased complexity on the nanoparticle surface leads to a shorter stability and usability. Based on the studies of Namazi et al. and Naeini et al., over time, citric acid and PEG are forming a self-aggregating system in our samples [7,15]. These dendrimers would offer a protective layer for PBNPs while encapsulating them along with the fluorescent dye MB. Our results suggest that further improvement would be necessary to synthesize a more stable PBPN platform for multimodal use as, e.g., MR-CA, radiotracer or therapeutic functions.

FTIR spectroscopy reveals the lengths of the bonds, the molecular mass changes, and the band-order of molecules. Hypsochromic shifts (shifts to lower wavenumbers) suggest that the mass of the corresponding molecule has been increased; the frequency of vibration is inversely proportional to mass of vibrating molecule. In addition, the absorption bands near 3415 and 1610 cm^−1^ refer to the O–H stretching mode and H–O–H bending mode, respectively, indicating the presence of interstitial water in the samples (Appendix A) [45,46,47].

In medical imaging, the safety and discomfort of patients are important; therefore, a significantly faster and safer method used for lengthy and repeated procedures could make a huge impact on both diagnostics and monitoring. Optical methods are peculiarly useful for visualizing soft tissues, which can be easily differentiated from each other based on their light-absorbing and -scattering capabilities. The dual modalities allow better chances for inspection, based on different kinds of imaging modalities, e.g., MRI and OI. Finally, if an acute intervention is required, the need of repeated CA administration could be preventable, if the half-life (and clearance) of the nanoteragnostic agent is adequate.

The total workings of the immune system play a significant role in the distribution, metabolism, and elimination of nanoparticles. As Diao et at. and Hong et al. used C57BL/6 mice of IR imaging, we also decided to test our nanosystem in C57BL/6 mice that have an intact immune system as opposed to immunocompromised nude mice [48,49]. This might have made our imaging signals more scattered, but increased the validity of the model towards, eventually, clinical translational studies.

The elimination and excretion of nanoparticles is, however, poorly investigated. Due to its mainly oral administration, only a few studies have been published data regarding I.V. PB administration. As far as we know, ultrasmall PBNPs tend to be excreted via glomerular pathways, while after oral administration, PBPNs are excreted via fecal and urine routes [13,50,51]. The few available reports dealing with biodistribution and elimination of PBNPs after intravenous injection are not conclusive. It would seem, though, that in those studies, kidney excretion was not relevant despite smaller PBNP deposits in the mesangium and the peritubular vessels of [13,15]. Our findings with the new coating and dye-adsorbed-modified PBNPs suggest that both kidney and biliary elimination routes play a major role in the excretion of this nanoparticle system. Shortly after iv. administration, the particles could be found in the urinary tract, showing an enhanced fluorescent signal 3 h after the admission. This, along with our former data obtained with irreversibly ^201^Tl-isotope-labelled PBNPs passing through the kidney to the urinary excretion route, suggests a portion- of kidney excretion via fenestrae of the kidney endothelium and podocytes, especially in our measured hydrodynamic size range below 40 nm [13].

In order to account for eventual dissociation of the MB dye from PBNP surfaces, we compare the known elimination routes and dynamics of MB dye solution to our MB-labelled PBNPs. Peter et al. reported that while MB alone accumulates in the brain and bile, mainly urinary excretion is significant 1 h after IV administration. The maximal concentration during urinary excretion was reached 2–4 h after administration [52]. In this context, our results suggest that MB stayed bound to PBNPs as both renal clearances are prolonged while the PEG shell alone did not significantly increase the renal excretion speed. The fluorescent intensity in the heart, liver, and spleen suggest a prolonged circulation time. Significant uptake can be observed in the lungs (501% ± 85%; 2 h post injection 156% ± 60%) and spleen (163% ± 18%), while a slight increase in fluorescent intensity in the intestines (107% ± 5%), along with the urinary excretion (272% ± 32%; 3 h post injection 177% ± 20%) (percent values based on normalized fluorescent intensities preinjection ± SD). These results suggest that the MB and PBNPs were connected, and their clearance followed the both characteristic pathways for MB and PBNP; furthermore, the PEG shell alone did not promote the renal excretion significantly. The fluorescent intensity in the heart, liver, and spleen suggest a prolonged circulation time (Appendix A).

The ex vivo images confirm that the application of PEG led to an increased biological half-life and slowed excretion rate, both biliary and urinary, compared to previous studies [53,54,55]. Even if other studies emphasize the connection between filtration and negative surface charge and larger particle size, PEGs are known as rather neutrally charged stabilizers. According to Liu et al. and Souris et al., complete clearance of PEG-modified nanoparticles last for 2 months, and thus, show a tendency of hepatic and renal accumulation causing local inflammation and necrotizing tissue [56,57].

Chen et al. investigated the acute and subchronic toxicological properties of Prussian blue nanoparticles after exposure of mice. PBNPs accumulated mainly in the spleen and liver. They found that, however, PBNPs induced acute damage in the liver (based on the liver functions), the long-term effects of PBNP-treatment cannot be called negative. All monitored parameters returned to normal levels, 60 days after the first IV PBNP administration [35]. Our results also confirm the claims of Chen et al.; in the ex vivo images (Figure 4), PBNPs were present in the spleen and the liver; biliary functions resulted in PBNP presence in the gastrointestinal tract.

The nontoxic properties of PBNPs were also confirmed by the study of Liang et al. Furthermore, they evince that no significant or fatal injury can be attributed to PBNP treatment, neither on histological nor on macroscopic matter. The vital organs were unharmed, no injuries were present on cellular levels 2 weeks and 4 weeks after the treatment [58].

Although fluorescent imaging capabilities of PBNPs were demonstrated in this work, further optimization and development of the platform is essential before any clinical implementation. Based on our work, the scale-up of PBNP-based CA is yet to be done while long-term studies for usability and effects in vivo are needed before human studies, too.

Our results suggest the use of PBNPs as a platform for the development of new-generation multimodal CAs, for OI and MRI imaging as well as therapeutic purposes. The biocompatible shell of the NPs is easy to modify due to the carboxylic groups, which would allow the cell or tissue targeted delivery. The targetable magnetic properties of the NPs would allow their theragnostic use, too, with proper selection of effector payload such as antibodies; thus, allowing a new type of topical infection-control method against bacterial and viral infections.

Such a NP system would act as CA in vivo as well as as a therapeutic agent for photothermal, isotope-based, biological or stem cell therapies, opening new aspects in the future of medicine.

## 5. Conclusions

In this work, we demonstrated a synthesis and modification methodology to biocompatible stealth fluorescent PBNPs. We created a novel nanoparticulate contrast material for the classical methodology of fluorescent measurements, e.g., using FOBI. Our results suggest that the stealth-liposome-like platform is suitable for fluorescent imaging.

We investigated the PB nanoobjects in vivo after intravenous admission and examined the elimination routes of these particles. We described the hepatobiliary and renal uptake and excretion to be both important in our system with in vivo intravenous application.

With both PEGylation and fluorescent labelling PB nanoparticles can be conferred with advanced technological properties to offer a finely tuned platform for clinical application after further development. We aim towards clinical translation hence the application of MB and NIR wavelength usage too. As fluorescent detection and imaging data in a suboptimal in vivo system are more convincing to extend towards human clinical trials, the use of standard and more clinically valid Black mice might have been a better choice for validity and detection studies than transparent but less clinically translatable nude mouse study.

## Figures and Tables

**Figure 1 nanomaterials-10-01732-f001:**
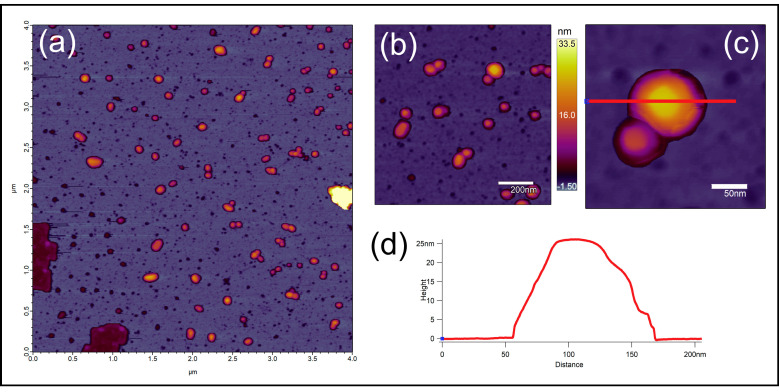
(**a**) Height-contrast AFM image of PBNPs on the mica surface (scale bar = 4 µm × 4 µm). (**b**) Modified PBNPs on the mica surface; height-contrast AFM image (scale bar = 200 nm). (**c**) A modified PBNP nanoparticle (scale bar = 50 nm). (**d**) The cross-section graph of (**c**) PBNP nanoparticle (abscissa = 0–200 nm; ordinate = 0–25 nm). AFM: atomic force microscopy; PBNPs: Prussian blue nanoparticles.

**Figure 2 nanomaterials-10-01732-f002:**
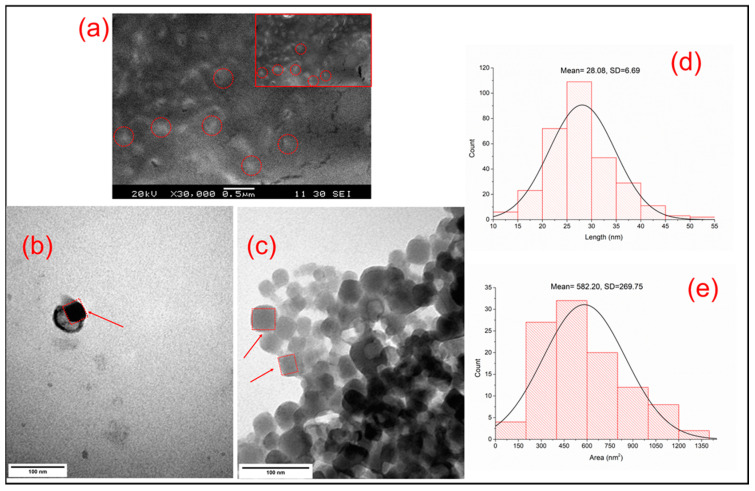
(**a**) The SEM micrograph of modified PBNPs (magnification: 30,000x, scale bar = 0.5 µm); the PBNPs are barely visible between the pores of the double-sided carbon tape (section of (**a**); magnification: 30,000×). (**b**–**c**) The TEM micrograph of modified PBNPs (magnification = 250,000x; scale bar = 100 nm). (**d**) The mean length of PBNPs, determined by TEM images. (**e**) The average area of the modified PBNPs according to TEM images. SEM: scanning electron microscopy; PBNPs: Prussian blue nanoparticles; TEM transmission electron microscopy.

**Figure 3 nanomaterials-10-01732-f003:**
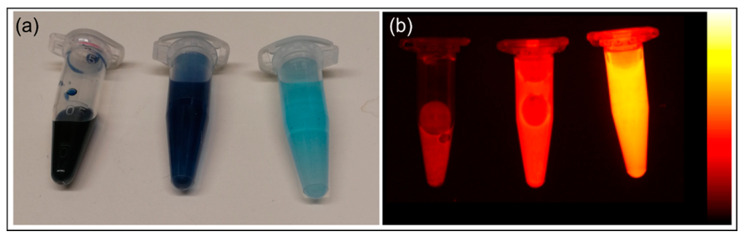
(**a**) PBNP-MB@PEG6000_100μL (stock solution × and 20× dilution left to right). (**b**) Fluorescent signal of PBNP-MB@PEG6000_100μL dilutions in FOBI. PBNP-MB@PEG6000_100μL: Methylene blue-labelled (100 μL Methylene blue) PEG 6000 stabilized Prussian blue nanoparticles; FOBI: Fluorescent Organism Bioimaging Instrument.

**Figure 4 nanomaterials-10-01732-f004:**
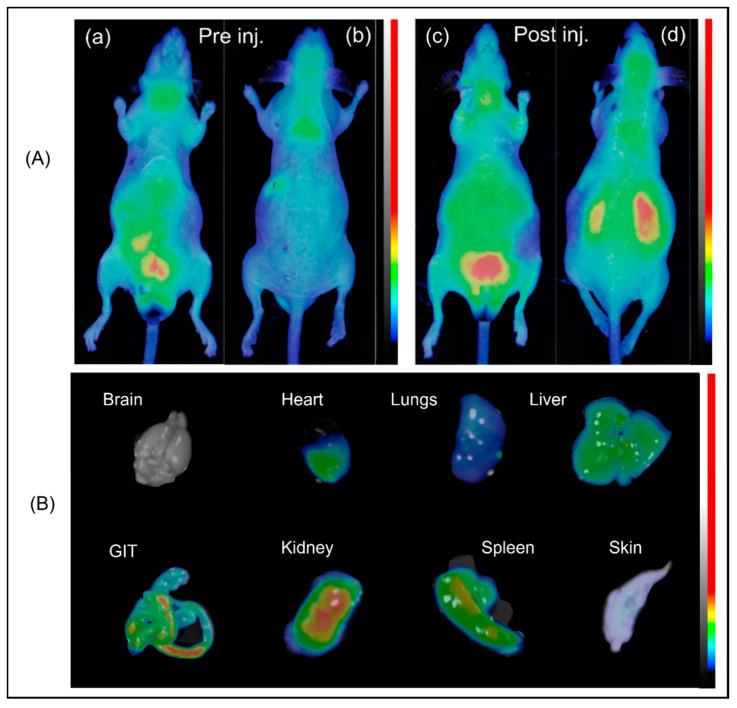
(**A**) (**a**,**b**) C57BL/6 mice before intravenous treatment from anterior and posterior planes. (**c**,**d**) C57BL/6 mice pre-PBNP treatment (anterior and posterior planes). (**a**) anterior, preinjection; (**b**) posterior, preinjection; (**c**) anterior, postinjection; and (**d**) posterior, postinjection. Color scale equals mean fluorescent intensity (MFI) in arbitrary units (A.U.) Figure 4 (**B**) ex vivo images, 3 h postinjection.

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
