# Peer review of "Fluorescent, Prussian Blue-Based Biocompatible Nanoparticle System for Multimodal Imaging Contrast"

_nanomaterials, 2020, doi:10.3390/nano10091732_

Round 1

Reviewer 1 Report

The authors describe a new Pussian blue nanosystem functionalized with fluorescent molecules for multimodal imaging applications. The topic of the work is interesting for the scientific community and the biological results (described in the last section of the document) are very encouraging to study and use this type of inorganic sample for diagnostic applications. For these reasons, the manuscript is suitable for publication after a revision of the chemical section that suffers from some problems, reported here:

1) The chemical decoration of PB nanoparticles with methylene blue (MB) molecules is not well specified and demonstrated in the paper. A clear indication of the chemical interactions should be reported in the text (I assume that ionic interactions occur between the positive MB groups and the negative charges of the citrate molecules). In this way, it is possible to detect a change in the surface charge (by analyzing the Z potential) of the PB nanoparticles after the MB labeling.

2) In the X-ray diffraction section, a comparison of the x-ray profiles of modified PB nanoparticles, the pure PEG used for the functionalization and the MB can help to understand the interpretation of the peaks in the Fig. 3B.

3) IR section: 1) Please enter the assignment of the most important peaks in the IR figures. 2) The band at 494 cm-1 attributed to the Fe-CN-Fe stretching is not detectable in these spectra. The background of the spectra is of poor quality. Probably, this vibration mode can be better detected by FT-Raman spectroscopy. 3) The IR spectrum of MB is not informative: two main absorptions at 3300 and 1630 cm-1 are visible and generally assigned to the stretching and bending modes of adsorbed water. The peaks attributed to the aromatic and the methyl groups of the molecule are not detectable. The authors are invited to repeat the experiment, after complete dehydration of the sample.

4) Do you have any indication of possible changes in the luminescent properties of the dye molecules after confinement to the PB surface? The reduction of dye aggregates with a clear improvement in quantum efficiency is often observed for luminescent molecules attached to the surface of inorganic nanoparticles.

Author Response

 Response to Reviewer 1 Comments

  • The chemical decoration of PB nanoparticles with methylene blue (MB) molecules is not well specified and demonstrated in the paper. A clear indication of the chemical interactions should be reported in the text (I assume that ionic interactions occur between the positive MB groups and the negative charges of the citrate molecules). In this way, it is possible to detect a change in the surface charge (by analyzing the Z potential) of the PB nanoparticles after the MB labeling.

Surface charge measurements have been concluded - they suggest, a small decrease in the Zeta potential after MB labelling. (Mean value before <-30 mV ; after the labelling <-25 mV). Also, additional surface charge measurements were concluded, on older samples. Their Zeta-potential did not exceed -20 mV. Therefore, it is possible, that the decreased surface charge is the result of the ionic interactions between the dye and the citrate surface. Nevertheless, we must not forget, that the pH of the used buffer (Phosphate buffer saline pH 6.8) could also result a decreased Zeta-potential.

  • In the X-ray diffraction section, a comparison of the x-ray profiles of modified PB nanoparticles, the pure PEG used for the functionalization and the MB can help to understand the interpretation of the peaks in the Fig. 3B.

While the main focus of our work was on the imaging capabilities of the fluorescently labelled modified PBNP system, XRD data and FT-IR owns smaller reputation. In this context, the data of these measurements was replaced tot he Supplemental Information. However, peaks of MB are presented in several work, (Ref. https://doi.org/10.1155/2013/923739); (Ref. https://www.researchgate.net/publication/281862069_Structural_investigation_by_X-ray_diffraction_analysis_on_the_behaviour_of_methylene_blue_towards_transition_metal_anions) as well as PEG 6000 was widely investigated and measured via XRD (Ref. https://www.researchgate.net/publication/341458921_Solid_dispersions_of_fenofibrate_Comparision_of_natural_and_synthetic_carriershttps://doi.org/10.1002/app.43027)

According to these References, 2Θ = 18.51, 23.13, 31.79, 35.37 peaks are related to MB, while 2θ = 26.090, 36.160, and 39.680 are corresponding peaks of PEG6000.

  • IR section: 1) Please enter the assignment of the most important peaks in the IR figures. 2) The band at 494 cm-1 attributed to the Fe-CN-Fe stretching is not detectable in these spectra. The background of the spectra is of poor quality. Probably, this vibration mode can be better detected by FT-Raman spectroscopy. 3) The IR spectrum of MB is not informative: two main absorptions at 3300 and 1630 cm-1 are visible and generally assigned to the stretching and bending modes of adsorbed water. The peaks attributed to the aromatic and the methyl groups of the molecule are not detectable. The authors are invited to repeat the experiment, after complete dehydration of the sample.

We’d like to thank the Refree to highlight the shortcomings of FT-IR figure. We have repeated the FTIR measurements, after a complete dehydration. The samples were freeze-dried, than were measured in powder form, as we previosuly described. Also, measurements in D2O were concluded, the results are in the Supplemental Information enclosed. We’ve considered 2), the FTIR specra were updated accordingly. Aromatic groups and methly groups are visible. Aromatic groups and methly groups are visible at 2807 and 2710 cm-1 represent the stretching vibration of -CH- and -CH3, aromatic vibrations are present from 1601-1372 cm-1, while the vibrations of the C=C skeleton of the dye are present at 1171 cm-1 (SI Figure S8 (b)). On the other hand, Figure  S8 (d) has a visible vibration band around 1170 cm-1, which could represent the C=C skeleton of the aromatic rings. Also, signs of C≡N bands are present at 2090 cm-1, and the board band of -OH around 3500 cm-1 and -C-O stretching around 1200 cm-1. These vibration and stretching bands can be related to both citric acid and PEG, as well was PB itself.

  • Do you have any indication of possible changes in the luminescent properties of the dye molecules after confinement to the PB surface? The reduction of dye aggregates with a clear improvement in quantum efficiency is often observed for luminescent molecules attached to the surface of inorganic nanoparticles.

This topic was already investigated by Sarmah et al. (Ref. https://doi.org/10.1021/acssuschemeng.6b01673).They synthesized mesoporous Fe-containing nanomaterials for photocatalic application. In this case, Fe-ions (Fe(ox)-Fe3O4) were present in the solutions, acting as photocatalists, which could lead the degradation of MB moleucles. In our case, the Fe-ions were in strong bond with -CN groups, the number of free Fe-ions was negligible. They also investigated the effect of pH on the degradation of MB. Around pH~7, the degradation they found was on the lowest level, compared to other pH values. The degradation was also catalised by visible light, UV and LED light sources. To overcome this issue, our samples were stored in dark places, the dialysis in PBS was also made under dark, controlled conditions.

We assume, the luminescent changes of MB absorbed onto PBNP surface are least possible. Quenching or energy transfer between PBPNs and MB are unlikely, although their emissions spectra slightly overlap. However, PBNPs are inorganic molecules with complex charge-transfer systems, as well as MBs are (with their delocalized electrons), the surface between PBNPs and MBs is sealed with the citric acid (as it is a capping agent). In fact, we assume citric acid makes it possible, MB to connect to the PBNPs.

Reviewer 2 Report

Title: Fluorescent, prussian blue-based biocompatible nanoparticles system for multi-modal imaging

This manuscript should be rejected because of many problems as follows

1/ This manuscript has no significance because of wrong methodology, failed experiments, and missing of a lot of necessary data for a fluorescent material.

2/ Methods for experiments of “fluorescent labeling” “pegylation” are wrong from beginning steps.

  • Mixing 2 pigments as Prussian blue (PB) and methylene blue (MB) together is leading to labeling MB to PB? How is reaction? What is conjugation here? How many percentage of MB labeled on PB? What is data to confirm the reaction?
  • Why does MB use in this work? since MB is easily oxidized or reduced in the presence of Fe3+/Fe2+ of PB and then becomes colorless à MB is a wrong choice
  • PEG has no chemical reactions with PBMB in this work, therefore PEG can’t prevent leakage of MB dye.
  • Adding EDC to PEG and PBMB for conjugation is a stupid step, because EDC is used for conjugation between carboxyl and amine groups, so in this work using EDC is wrong

3/ In fact that the experiment with MB is also failed because

  • Data in Fig. 1 and Fig. 2 is contrasted
  • Data in Fig. 3 and Fig. 4 have no meaning because of no reaction here

4/ Working with a fluorescent-based material for imaging but there is no data for absorption, emission spectra, fluorescent quantum yield, life time, thermostability, etc. Without these information how authors can work for bio-imaging?

5/ In fact that fluorescent quantum yield of MB is very low, therefore, fluorescence emission in Fig. 5 can’t occur for MB. Moreover, authors can’t show graphs for absorption, excitation and emission wavelengths of PBMB, thereby photo in Fig. 5 could be of another dye, not MB.

6/ Text in page 12-14 show a poor knowledge and wrong explanation in chemistry about chemical bindings of PB, MB, PEG, EDC. Citing other references for explanation but authors don’t understand what have done in these references.

7/ Toxicity of any material should be tested before using on living organisms

8/ Photos and caption in Fig. 6 don’t fit together

9/ Experiments of using Eosine Y and Rhodamin B are presented as failed experiments, why are they mentioned in this work?

10/ Something in supplement information is not related to this work

Author Response

Response to Reviewer 2 Comments

1/ This manuscript has no significance because of wrong methodology, failed experiments, and missing of a lot of necessary data for a fluorescent material.

1/ We strongly refuse the Referee`s pre-judgment based on the apparent lack of his knowledge or lack of due referencing of the field of nanoparticle production and pegylation. Ample literature describes theoretically and practically that the same as our methodology applied in thousands of other experiments published accross the full spectrum of literature and in industrial scale, has yielded the same results with "stealth" nanoparticles most frequently achieved using PEG. We have been one of the pioneers of the field of imaging PB nanoparticles, based on Shoukouhimehr`s work.

2/ Methods for experiments of “fluorescent labeling” “pegylation” are wrong from beginning steps.

References for labelling mesoporous nanoparticles with fluorescent dye:

Ref. https://doi.org/10.1021/jp303199s

Ref. https://doi.org/10.1039/C3NR00119A

Ref. https://doi.org/10.1016/j.biomaterials.2011.06.019

We would gladly recommend checking the following references on the topic of the pegylation:  (Ref. https://doi.org/10.1016/j.biomaterials.2004.04.014; https://doi.org/10.1016/j.nano.2009.11.008).

Mixing 2 pigments as Prussian blue (PB) and methylene blue (MB) together is leading to labeling MB to PB?

Yes, MB will adsorb to the mesoporous surface of PBNPs.

How is reaction? What is conjugation here? How many percentage of MB labeled on PB? What is data to confirm the reaction?

Surface charge measurements have been concluded - they suggest, a small decrease in the Zeta potential after MB labelling. (Mean value before <-30 mV; after the labelling <-25 mV). Also, additional surface charge measurements were concluded, even after 6 weeks. In these samples, Zeta-potential did not exceed -20 mV. Therefore, it is possible, that the decreased surface charge is the result of the ionic interactions between the dye and the citrate surface. Nevertheless, we must not forget, that the pH of the used buffer (Phosphate buffer saline pH 6.8) could also result a decreased Zeta-potential.

Why does MB use in this work? since MB is easily oxidized or reduced in the presence of Fe3+/Fe2+ of PB and then becomes colorless à MB is a wrong choice

Our previous experiments with other fluorescent dyes suggested, that we should consider the excitation and emission spectra of the dyes. MB has an excitation and emission max. over 600 nm, namely in the NIR window, which makes it a promising candidate. Furhermore, MB was reported to be the most table, when Fe ions are present, at a pH of 7 (the used buffer solution, PBS pH~6.8), which suggests the best possible outcome of the experiments.

We would like to remind the referee that methylene blue has been used in clinical image guided surgery in the past decade as a fluorophore. We also remind the referee to kindly remedy his ignorance in clinically relevant developments, because all materials we applied in our study by design, should have been clinically approved molecules. Our program is to develop nanoparticles with clinical relevance, to expedite translation of our nanoparticle systems for the benefit of clinical patients not to please the self-confident opinion of any one referee.

Looking up clinical practical literature, there is a vast amount of published procedures using MB as an FDA-approved fluorescent tracer. For example, even developers of new fluorescent dyes for clinical use state: "There are only two clinically available NIR fluorophores, indocyanine green (ICG) and methylene blue (MB), that support the image-guided surgery." Ref. 10.4068/cmj.2017.53.2.95 . or John Frangioni`s studies like in: https://www.ncbi.nlm.nih.gov/pmc/articles/PMC4644113/ and in this study, where, contrary to the referee`s opinion MB NIR fluorescence has been proven to find more tumor foci than SPECT itself. REF https://pubmed.ncbi.nlm.nih.gov/23720199/

MB has been the right choice for the above reason of serving the public.

Further details of the adsorption of MB to porous surface nanoparticles are found in REF https://pubmed.ncbi.nlm.nih.gov/32668550/. Kinetics and modeling of the sorption process to other high surface porous particulate systems are exemplified in REF https://pubmed.ncbi.nlm.nih.gov/32566358/

PEG has no chemical reactions with PBMB in this work, therefore PEG can’t prevent leakage of MB dye.

As we mentioned, PEG and citrate (citric acid) was reported to form dendrimer structures. These could also act as a “protective” layer, therefore, the function of peg as capping agent - to prevent MB dye-leakage is clear. We have evaluated our data based on our leakage studies, and we base our statement on these facts.

Adding EDC to PEG and PBMB for conjugation is a stupid step, because EDC is used for conjugation between carboxyl and amine groups, so in this work using EDC is wrong

EDC was accidentally left in the manuscript, while it was used in another work similar to this topic.

(Regardless, theoretically, knowing the mechanism of the EDC reaction, the reviewer is advised to consider the following: Basically, EDC creates an activated carbon (carbocation), which, in later reactions reacts with amine groups. Activated carbon increases the negativity of -COOH, which suggests some possible electrostatic reaction between positive MB molecules and negative citrate (-COOH) groups. Furthermore, if we assume, that the small amount of EDC, which we added to the solutions did not activate all of the free -COOH groups of citric acid, there will be free -COOH groups that are able to form dendrimers with the PEG added to the solutions. Ref. Hermanson, Greg T. (2013). Bioconjugate Techniques, 3rd edition.)

Maybe the referee should inform the manufacturers of PEG-coated systems and the Notified Bodies allowing the former to be used in clinical applications that their programs are just to vanish in the air. We refer to (ClinicalTrials.gov Identifier: NCT03531827)

3/ In fact that the experiment with MB is also failed because

Data in Fig. 1 and Fig. 2 is contrasted

Data in Fig. 3 and Fig. 4 have no meaning because of no reaction here

3/ Figure. 1 and figure 2. were capture with the built-in software (IgorPro) of the Cypher AFM and the SEM and TEM microscopes. The images were not modified. The applied filter of the figure 1. is called OragneVioletYellow2 and was originally created to enhance the color-based differences for the viewers on height-contrast AFM images.

Per comments of Reviewer 1, we considered repeating parts of FTIR and XRD measurements, the results are enclosed in the new version of the manuscript. Having any questions, please feel free to contact us. 

4/ Working with a fluorescent-based material for imaging but there is no data for absorption, emission spectra, fluorescent quantum yield, life time, thermostability, etc. Without these information how authors can work for bio-imaging?

We would like to remind the Referee, the method we used for the imaging was not a quantitative method, as fluorescent two-photon microscopy or other optical fluorescent methods. FOBI imaging does not require such specific data; a simple calibration to the fluorescent dye used is enough for the imaging, therefore this method can only be called semi-quantitaive.  It is though possible, to capture quantitative images, using this device and method, however our main goal was to take a snapshot, if our sample does provide fluorescent signal in the living organism and to investigate, if this fluorescent signal provides us further information on the behavior of our material in the living organism.

5/ In fact that fluorescent quantum yield of MB is very low, therefore, fluorescence emission in Fig. 5 can’t occur for MB. Moreover, authors can’t show graphs for absorption, excitation and emission wavelengths of PBMB, thereby photo in Fig. 5 could be of another dye, not MB.

5/ To be clear, is the Referee saying that the use of FOBI device with any kind of fluorescent material would require specific measurements of fluorescence (excitation, emission and absorption), therefore, in this sense, our results do not reflect reality? Furthermore, other authors, who, in their work, used methylene blue as fluorescent imaging agent, must have been also misinterpreted their results (or basically just published a fraud?)(Ref. https://doi.org/10.1016/j.surg.2009.12.003; https://doi.org/10.1016/j.juro.2013.02.3187; https://doi.org/10.1002/jso.25105). If so, please consider contacting other authors about the shortcomings of their work. Otherwise, saying, that the fluorescent emission and quantum yield of a dye is low, does not mean it is not present. As we know, any kind of fluorescence is concentration dependent, self-absorption and quenching can occur in concentrated solutions. Figure 5 (figure 3. in the updated version) show different dilutions of MB, which in concentrated solution, self-absorption is present. As it gets diluted, fluorescent emission happens – as figure 3 shows.

(Regardless, quenching or energy transfer between PBPNs and MB are unlikely, although their emissions spectra slightly overlap. However, PBNPs are inorganic molecules with complex charge-transfer systems, as well as MBs are (with their delocalized electrons), the surface between PBNPs and MBs is sealed with the citric acid (as it is a capping agent). In fact, we assume citric acid makes it possible, MB to connect to the PBNPs.)

Any direction you can provide on this matter is greatly appreciated to help the public better understand our work. Going forward in the interest of transparency, please also feel free to request samples, if you are to repeat the measurements, presented in our manuscript. 

6/ Text in page 12-14 show a poor knowledge and wrong explanation in chemistry about chemical bindings of PB, MB, PEG, EDC. Citing other references for explanation but authors don’t understand what have done in these references.

We kindly ask the Referee to explain which pages he or she is referring to. In our manuscript pages 12-14 do not explain the chemistry of either PB, MB, PEG or EDC. Page 12-14 sums up our results of in vitro fluorescent measurements (3.2.4.), Stability measurements of PBNP-MB@PEG nanoparticles (3.2.5.) and the in vivo and ex vivo measurements (3.3.), while page 14 gives an in depth history of PBNPs and different imaging modalities, including fluorescent imaging. (Please note, that the updated manuscript does not contain the corresponding pages as they were previously, pages 12-14 you are referring to are equal to pages 9-11 in the updated version.)

7/ Toxicity of any material should be tested before using on living organisms

You seemed to have missed something in the references. (Ref. https://doi.org/10.1177/0960327115622258)

The preliminary results on this topic are listed here. We have already investigated the toxicity of the PBNPs on cellular level, furthermore, Thallium labelled PBNPs were also tested in mice (Ref. https://doi.org/10.1155/2018/2023604). The method used for the synthesis of PBNPs based on Shoukouhimehr`s work (Ref. https://doi.org/10.1016/j.inoche.2009.10.015). In terms of relative safety, based on literature data (Ref. https://www.ncbi.nlm.nih.gov/pmc/articles/PMC3087269/; Ref. http://accessmedicine.mhmedical.com/content.aspx?bookid=391&sectionid=42070020; Ref. http://www.inchem.org/documents/jecfa/jecmono/v14je19.htm), the amount of PB, MB and PEG6000 we applied was well below both human and rodent toxic doses for both MB and PEG6000.

8/ Photos and caption in Fig. 6 don’t fit together

8/ There must be a misunderstanding about fig 6. This figure has two parts, the upper part (Fig. 6/(1)) is related to the in vivo measurements after the PBNP-MB@PEG6000 administration, while the lower part (Fig. 6/(2)) shows the ex vivo measurement data.

9/ Experiments of using Eosine Y and Rhodamin B are presented as failed experiments, why are they mentioned in this work?

10/ Something in supplement information is not related to this work

9-10/ The main nature of our work was to create a fluorescently labelled PBNP system. To fulfill our goal, it was needed to try not only one but several (at least 2) fluorescent dyes - of which only one was chosen, while the others were considered “unsuccessful” attempts – as the Referee perceives.

The fact that the reviewer points out here that other experiments were unsuccessful, while labelling with MB resulted fluorescently labeled particles and in a stable NP system ( supported by measurement data), refutes itself and gives the impression that the whole study is a total failure without work. Also, the referee stated that MB can’t be used for fluorescent imaging, due to its limited capabilities, however the cited references support the authors results.

Even though, a failed experiment should be considered as result, so, in the spirit of striving for excellence, we wanted to give the audience a brighter image to reveal every detail of our whole work. All in all, communicating all data related even to the smallest step of this work was in the interest of transparency.

Round 2

Reviewer 1 Report

On the base of the answers, the manuscript is now suitable in the present form.

Author Response

We'd like to express our gratitude to the reviewer for the favourable and well based decision. 

Kind regards, 

László Forgách